# AVRT: Audio-Visual Reasoning Transfer through Single-Modality Teachers

## Abstract

While recent advances in reasoning models have shown remarkable progress in text-based domains, the development of effective reasoning capabilities in multimodal settings, particularly audio-visual, remains still a challenge, mainly because of the limited availability of high-quality reasoning data in target multimodal combinations. To address this problem we introduce AVRT, a novel framework that generates high-quality audio-visual reasoning data by distilling knowledge from specialized single-modality teachers. To this end, we generate high-quality reasoning traces via a vision-reasoning and an audio-reasoning teacher and merge the resulting traces with an LLM merger model. This enables a two stage training with a supervised fine-tuning of student models as cold start followed by a reinforcement learning. Our evaluation shows that the resulting models achieve superior performance on various datasets, i.a. OmniBench, DailyOmni, and MMAR, establishing a new pipeline for an effective training of audio-visual reasoning models. [1]

## 1 Introduction

Humans perceive the world by combining information from multiple modalities through diverse sensory inputs. With the wide availability of multimodal data, such as videos, multimodal understanding in general and audio-visual understanding in particular has drawn more and more interest from the research community. Recent advancements in this area, also in combination with large language models, have shown remarkable performance in audio-visual understanding (Cheng et al. (2024); Xu et al. (2025); Liu et al. (2025b); Comanici et al. (2025); OpenAI (2024)).

In parallel, the emergence of reasoning-capable language models has led to new capabilities with respect to the analysis and understanding of a given scenario, exemplified by OpenAI's o-series (Jaech et al. (2024)) and DeepSeek-R1 (DeepSeek-AI et al. (2025)). These advances have been significantly driven by reinforcement learning techniques (Shao et al. (2024)). These reasoning capabilities have successfully extended to inputs beyond text, such as vision-text models (Huang et al. (2025); Dong et al. (2025)) and audio-text models (Xie et al. (2025); Wen et al. (2025); Goel et al. (2025), demonstrating chain-of-thought capabilities within the respective modalities. However, audio-visual reasoning has not yet seen the same level of advancement as its single-modality counterparts, i.a. due to the challenge of integrating information and reasoning cues across different modalities at scale as well as the inherent lack of data for audio-visual reasoning. Existing approaches try to address this problem e.g. by generating reference reasoning chains from large foundation teacher models that were trained with all target modalities (Du et al. (2025)) or try to approach the problem by extending reinforcement learning formulations e.g. by improved credits assignment or by context summarization (Yang et al. (2025); Kulkarni & Fazli (2025)).

This paper proposes a new pipeline for Audio-Visual Reasoning Transfer (AVRT) based on single-modality teachers. Namely, we systematically combine reasoning chains from specialized single-modality reasoning models via a text-only LLM merger model to generate coherent multimodal reasoning that explicitly correlates information across audio and visual channels. By using an LLM as a merging interface between the teacher models and the resulting reasoning chain, we have the freedom to prompt the models in the format that they were optimized for, leading to high-quality

---

[1]All code, data, and checkpoints will be made available.

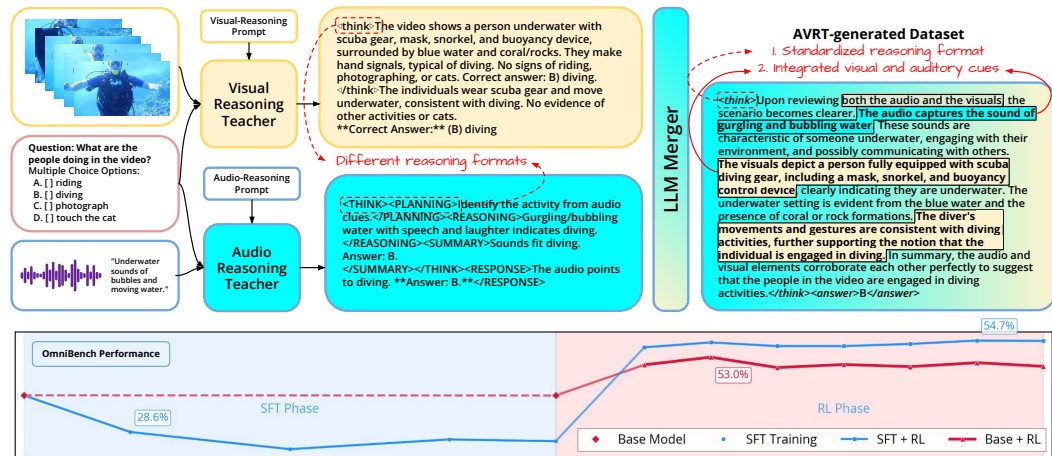

Figure 1: **Top:** Overview of the AVRT pipeline to generate reasoning chains that integrate visual and auditory cues. We first generate reasoning chains from single-modality teacher models allowing to prompt the models in the format that they were optimized for and, second, leverage an LLM merger as an interface between the teacher models and the resulting reasoning chain to aggregate the information and put it into the target format. **Bottom:** Zero-shot accuracy on OmniBench (Li et al. (2024)) during two-stage training on AVQA (Yang et al. (2022)): The SFT phase on the AVRT-generated reasoning chains serves as a "cold-start" stage for the model, yielding better performance during the RL training, when compared to the RL training alone.

modality-specific outputs. The following LLM merger then allows to aggregate the information and converting it into the target format. To this end, as shown in Fig. 1, AVRT extracts detailed chains of thought independently from specialized visual and audio teachers, then merges them with a text-only aggregator into a single cross-modal reasoning trace. These merged traces are then, in the first step, used to fine-tune a student audio-visual large language model in a cold-start manner via supervised fine-tuning (SFT) to distill the reasoning formatting and patterns, as well as to integrate multimodal reasoning into the student model. In a second step, the student model is trained following the GRPO (Shao et al. (2024)) objective.

We evaluate the proposed approach on four challenging datasets: OmniBench (Li et al. (2024)) and DailyOmni (Zhou et al. (2025)), as zero-shot vision-audio downstream datasets, AVQA Yang et al. (2022) as in-domain validation dataset, and MMAR (Ma et al. (2025)) as audio-only downstream task. Using only a 3B-parameter student, we achieve improved performance compared to other 3B audio-visual reasoning models and competitive performance to 7B models. Our ablation shows that the combination of reasoning traces of two different teacher models by a language-only merger model allows to improve audio-visual reasoning in the GRPO learning stage.

Our contribution can be summarized as follows: 1) We propose a novel method to aggregate multiple single-modality reasoning chains into integrated multi-modal reasoning traces, creating high-quality audio-visual reasoning data without expensive annotation. 2) We are the first to train a multimodal reasoner based on this type of composed reasoning data, demonstrating that cross-modal understanding can emerge from the combination of single-modality expertise. 3) We conduct an extensive evaluation on audio-visual benchmarks, achieving state-of-the-art results that compete with larger models through our approach.

## 2  RELATED WORK

**Audio-Visual Large Language Models.**   Audio-visual large language models have made constant progress in addressing challenging tasks in multimodal understanding. Early works like Meerkat (Chowdhury et al. (2024)) focus on fine-grained spatial and temporal grounding on five audio-visual tasks, introducing optimal transport-based modality alignment and cross-attention modules for audio-visual consistency. AVicuna (Tang et al. (2025)) proposes specifically targeting tem-

poral referential dialogue in untrimmed videos, introducing the Audio-Visual Tokens Interleaver for temporal alignment and developing comprehensive datasets like A5-222K and PU-VALOR. VideoLLaMA 2 (Cheng et al. (2024)) further advances spatial-temporal modeling through specialized Spatial-Temporal Convolution connectors and enhanced audio understanding via joint training, achieving state-of-the-art performance among open-source models. Rather than developing new models from scratch, PAVE (Liu et al. (2025a)) introduces a lightweight adaptation framework that extends existing video LLMs to handle diverse side-channel signals through efficient "patches" that add only 0.1% additional parameters. More recent work has moved toward comprehensive omni-modal capabilities, with Qwen2.5-Omni (Xu et al. (2025)) enabling end-to-end streaming multi-modal inputs and outputs through innovations like TMRoPE for synchronizing video with audio and the Thinker-Talker architecture for concurrent text and speech generation. Ola (Liu et al. (2025b)) pushes these frontiers further with progressive modality alignment training strategies that use video as a central bridge to connect modalities, achieving performance comparable to specialized models across image, video, and audio understanding tasks. Despite these advances, these existing approaches often struggle to effectively associate information across both modalities, lacking structured reasoning processes that can explicitly capture and leverage cross-modal dependencies and correlations for comprehensive multimodal understanding.

**Audio-Visual Reasoning.** The field of audio-visual reasoning with large language models has seen rapid advancements, focusing on enabling models to comprehend and reason over complex multi-modal inputs. AURELIA (Chowdhury et al. (2025)) presents a novel actor-critic based reasoning framework that distills structured, step-by-step reasoning into AVLLMs at test time without additional training. Daily-Omni (Zhou et al. (2025)) introduces a dedicated Audio-Visual QA benchmark of 684 videos focusing on temporally-aligned multimodal reasoning in daily life scenarios, accompanied by Daily-Omni-Agent, a training-free agent that utilizes an open-source visual language model (VLM), an audio language model (ALM) and an automatic speech recognition (ASR) model to establish a baseline for this benchmark. Recent work has increasingly leveraged reinforcement learning: EchoInk-R1 (Xing et al. (2025)) proposes a framework using Group Relative Policy Optimization (GRPO) for structured cross-modal reasoning, demonstrating emergent self-corrective reasoning behaviors. HumanOmniV2 (Yang et al. (2025)) addresses shortcut problems by requiring explicit context summarization before reasoning, introducing context and logical rewards alongside IntentBench for understanding human intentions. Omni-R1(Zhong et al. (2025)) tackles the trade-off between temporal coverage and spatial resolution through a two-system architecture with hierarchical rewards. Finally, AVATAR (Kulkarni & Fazli (2025)) presents an off-policy RL framework with Temporal Advantage Shaping (TAS) for improved credit assignment during planning and synthesis stages. Instead of mitigating data scarcity with increasingly complex RL frameworks, we present a more direct paradigm. We distill knowledge from specialized open models to construct a unified reasoning dataset, which then allows a simple supervised fine-tuning (SFT) approach, supplemented by minimal RL, to significantly outperform these computationally burdensome methods.

**Audio-Visual Datasets.** AVQA (Yang et al. (2022)) can be considered one of the foundational audio-visual QA datset with $57,335$ question-answer pairs from daily audio-visual activities requiring clues from both modalities. OmniInstruct (Li et al. (2024)) develops a comprehensive tri-modal reasoning dataset combining visual, audio, and textual resources, while MUSIC-AVQA (Li et al. (2022)) expands to musical performance with $45,867$ question-answer pairs across $9,288$ videos. AVQA-R1-6K (Xing et al. (2025)) provided a manually curated subset of OmniInstruct focusing on questions that were more likely to require audio-visual reasoning. While these datasets have advanced the field significantly, they primarily focus on question-answer pairs without providing explicit reasoning traces that demonstrate how models should integrate cross-modal information. Our AVRT approach directly addresses this gap through structured reasoning chain generation, as compared in Table 1.

## 3 METHODOLOGY

In this paper, we derive audio-visual reasoning chains based on existing audio-visual question-answering pairs as e.g. provided by AVQA (Yang et al. (2022)). In this section, we first discuss the generation of audio-visual reasoning chains in Sec. 3.1 and our training procedure to leverage this data to perform audio-visual question answering in Sec. 3.2.

Table 1: **Comparison of audio-visual question answering datasets.** Our dataset provides reasoning traces that explicitly demonstrate cross-modal integration between audio (A) and visual (V) modalities, addressing a key limitation in existing AVQA datasets which focus solely on question-answer pairs without intermediate reasoning steps. All datasets use multiple-choice questions with 4 options (MCQ-4). The number of QA pairs is reported in thousands (K).

| Dataset | Modalities | Answer Format | # QA pairs (K) |
|---|---|---|---|
| AVQA | A+V | Video + question + 4-way answer | 57.3 |
| OmniInstruct-v1 | A+I | Image + question + 4-way answer | 96.1 |
| MUSIC-AVQA | A+V | Video + question + 4-way answer (focused on music) | 45.9 |
| AVQA - R1 - 6K | A+I | Video + question + 4-way answer (subset of OmniInstruct) | 6.4 |
| **Ours** (generated by AVRT) | A+V | Video + question + 4-way answer + Reasoning chains incorporating audio and visual data | 19.2 |

### 3.1 CROSS-MODAL REASONING CHAIN GENERATION

Our data generation process consists of two main stages: a single-modality reasoning extraction step and a cross-modal aggregation step. Let a audio-visual question-answering data sample be denoted as $(X, Q)$, where $X$ is a video and $Q$ is a question. The video $X$ comprises both an audio stream $A$ and a visual stream $V$, such that $X = (A, V)$.

**Single-Modality Reasoning Extraction.** In the first stage, we generate modality-specific reasoning. We select specialized teacher models for the audio ($T_A$) and visual ($T_V$) modalities. For a given sample $(X, Q)$, we provide each teacher with the question and its corresponding modality. We use carefully crafted prompts, $P_A$ and $P_V$, to elicit detailed reasoning chains. The audio reasoning chain is generated as $R_A = T_A(Q, A, P_A)$, and the visual reasoning chain is $R_V = T_V(Q, V, P_V)$. These chains capture the unique characteristics and patterns of each modality.

**Cross-Modal Aggregation.** In the second stage, we perform cross-modal aggregation. We use a text-only large language model, $M_{agg}$, to merge the reasoning outputs. This model takes the reasoning chains from both modalities and the original question to produce a unified, cross-modal reasoning output: $R_{agg} = M_{agg}(Q, R_A, R_V)$. This aggregation step transforms the diverse reasoning formats into a uniform structure, correlating characteristics from both modalities and incorporating cross-modal signals and dependencies.

**Filtering.** A critical consideration in our pipeline is handling cases where one or both teacher models produce incorrect answers. For this work, we retain only samples where both modality-specific teachers generate correct responses. This filtering strategy ensures high-quality training data by avoiding the propagation of erroneous reasoning patterns that could introduce noise during cross-modal aggregation (Turpin et al. (2023); Xie (2024)).

### 3.2 TRAINING

**Stage 1: Supervised Fine-Tuning** We fine-tune the base model on the merged audio-visual reasoning chains using a standard autoregressive language loss. Given a training sample $(X, Q, R_{agg})$ where $X = (A, V)$ is the video with audio and visual streams, $Q$ is the question, and $R_{agg}$ is the aggregated reasoning chain, we optimize the cross-entropy loss:

$$\mathcal{L}_{SFT} = - \sum_{t=1}^{|R_{agg}|} \log p_\theta(r_t | X, Q, r_{<t}), \tag{1}$$

where $r_t$ represents the $t$-th token in the reasoning chain $R_{agg}$ and $\theta$ are the model parameters. The model learns to generate structured reasoning following the format established during cross-modal aggregation: `<think>...</think><answer>...</answer>`, where the thinking section contains the multimodal reasoning process and the answer section provides the final response.

**Stage 2: Reinforcement Learning** In a second step, we employ Group Relative Policy Optimization (GRPO)(Shao et al. (2024)). GRPO eliminates the need for explicit value function estimation by deriving advantage estimates through group-based comparisons of model outputs.

The GRPO training operates by sampling $G$ distinct responses $\{o_1, o_2, \ldots, o_G\}$ for each input question $q$ using the current policy $\pi_{\theta_{old}}$. Each response $o_i$ receives a scalar reward $r_i$ from our reward function. The advantage for response $o_i$ is computed by normalizing rewards within the group:

$$\hat{A}_{i,t} = \widetilde{r}_i = \frac{r_i - \text{mean}(\mathbf{r})}{\text{std}(\mathbf{r})}, \tag{2}$$

where this advantage $\widetilde{r}_i$ is applied uniformly across all tokens $t$ in response $o_i$.

Our reward function incorporates multiple components to ensure correctness and proper formatting:

$$r_i = R_{format}(o_i) + R_{acc}(o_i) \tag{3}$$

The reward consists of two components:

**(1) Format Reward ($R_{format}$):** A binary reward that verifies adherence to the our proposed reasoning format (`<think>...</think><answer>...</answer>`):

$$R_{format}(o_i) = \begin{cases} 1, & \text{if format is correct} \\ 0, & \text{otherwise} \end{cases} \tag{4}$$

**(2) Final Answer Accuracy ($R_{acc}$):** A simple string matching evaluation that compares the model's predicted answer choice against the ground truth label:

$$R_{acc}(o_i) = \begin{cases} 1, & \text{if answer is correct} \\ 0, & \text{otherwise} \end{cases} \tag{5}$$

The GRPO objective maximizes the following function:

$$\mathcal{J}_{GRPO}(\theta) = \mathbb{E}_{\mathcal{D}} \left[ \frac{1}{G} \sum_{i=1}^{G} \frac{1}{|o_i|} \sum_{t=1}^{|o_i|} \left\{ \min \left[ \rho_{i,t} \hat{A}_{i,t}, \right. \right. \right.$$
$$\left. \left. \left. \text{clip} \left( \rho_{i,t}, 1 - \epsilon, 1 + \epsilon \right) \hat{A}_{i,t} \right] - \beta \text{KL}_{i,t} \left[ \pi_\theta || \pi_{ref} \right] \right\} \right], \tag{6}$$

where $\rho_{i,t} = \frac{\pi_\theta(o_{i,t}|q,o_{i,<t})}{\pi_{\theta_{old}}(o_{i,t}|q,o_{i,<t})}$ represents the probability ratio between policies, $\epsilon$ controls the clipping range, and $\beta$ weights the KL divergence regularization term against a reference policy $\pi_{ref}$.

## 4 EXPERIMENTS

### 4.1 DATASETS

#### 4.1.1 AVRT-20K DATASET

To train our model, we introduce the AVRT-20K dataset, which is constructed using our proposed AVRT method on a subset of the AVQA dataset. We use Kimi-VL-Thinking (Team et al. (2025)) and Audio Flamingo 3 (*think*) (Goel et al. (2025)) as the single-modality teachers $T_V$ and $T_A$ respectively. These models were chosen due to their balance between achieving state-of-the-art results in their modalities, and generating descriptive reasoning chains. We use 10-second audio input and 8 uniformly-sampled video frames from each sample as the input for the audio and visual teacher, respectively. The full prompt templates used for each model can be found in the Appendix (A.2).
.

Table 2: Statistics of the AVRT dataset showing sample counts, quality metrics, and distribution of question types and answer options across training and validation splits.

| Metric | Train / Val | Question Type | Train / Val (%) |
|---|---|---|---|
| Total Samples | 18,279 / 945 | Which | 45.2 / 45.7 |
| Reasoning Format Compliance | 100.0% / 100.0% | Come From | 30.9 / 29.8 |
| Thinking Section Length (tokens) | $165.5 \pm 33.9$ / $163.4 \pm 32.5$ | Happening | 15.5 / 14.1 |
| Answer Section Length (tokens) | $1.0 \pm 0.0$ / $1.0 \pm 0.0$ | Where | 8.0 / 9.7 |
| Video and Audio Duration (sec) | $10.0 \pm 0.1$ / $10.0 \pm 0.2$ | Why | 0.2 / 0.4 |
| Primary Resolution | 1280×720 (62% / 43%) | Others | 0.2 / 0.3 |

**AVRT-20K Statistics.** Table 2 presents statistics for our AVRT-20K dataset. The final collection comprises $18,279$ training samples and $945$ validation samples, all extracted from the original AVQA dataset. All samples achieve $100\%$ reasoning format compliance, ensuring consistent structure across the dataset. The thinking sections contain an average of $165.5 \pm 33.9$ tokens in the training set and $163.4 \pm 32.5$ tokens in the validation set, while answer sections are consistently single tokens ($1.0 \pm 0.0$), corresponding to the (A, B, C, D) options format. Videos maintain uniform duration of approximately 10 seconds ($10.0 \pm 0.1$ for training, $10.0 \pm 0.2$ for validation), with the primary resolution being 1280×720 (62% of training samples and 43% of validation samples).

The distribution of question types closely mirrors that of the original AVQA dataset, with "Which" questions being most prevalent (45.2% in training), followed by "Come From" (30.9%), "Happening" (15.5%), and "Where" (8.0%) questions. This similarity demonstrates that our random sampling successfully encompasses the distributional characteristics of the original dataset, ensuring our subset maintains representativeness across different reasoning types and question categories.

### 4.1.2 BENCHMARK DATASETS

We evaluate our model on four representative datasets that span different modality combinations to comprehensively assess cross-modal reasoning capabilities: DailyOmni (video+audio), OmniBench (image+audio), AVQA (video+audio, in-domain) and MMAR (audio-only) to examine potential overfitting on our training distribution.

**DailyOmni** (Zhou et al. (2025)) is a benchmark for evaluating multimodal large language models on real-life audio-visual scenarios that require joint reasoning across video, audio, and textual information. The dataset contains $684$ videos and $1,197$ question-answer pairs ($550$ from 60-second videos, $647$ from 30-second videos) covering all $11$ YouTube categories to ensure diversity of topics, styles, and acoustic environments. The questions are deliberately designed to force integration of modalities, moving beyond simple perception to complex reasoning tasks that require understanding of concurrent multimodal phenomena including speech, music, and environmental sounds.

**OmniBench** (Li et al. (2024)) was designed to evaluate large language models' ability to integrate image, audio, and text inputs for cross-modal reasoning. The benchmark contains $1,142$ question-answer pairs organized into $8$ task categories: Action & Activity, Story Description, Plot Inference, Object Identification & Description, Contextual & Environmental, Identity & Relationship, Text & Symbols, and Count & Quantity. Each sample includes multiple-choice questions with corresponding image and audio content, with audio clips averaging 9.22 seconds in duration.

**AVQA** (Yang et al. (2022)) is a large-scale benchmark containing $57,015$ question-answer pairs across $45,867$ videos designed to evaluate models' ability to reason over both audio and visual content. The dataset features high-quality manual annotations and questions that specifically require integration of both modalities, making it well-suited for evaluating genuine cross-modal reasoning capabilities rather than single-modality shortcuts.

**MMAR** (Ma et al. (2025)) is an audio-only reasoning benchmark designed to evaluate models' ability to perform complex reasoning tasks using solely auditory information. We include MMAR to assess how well our cross-modal training approach transfers to single-modality audio reasoning scenarios. The benchmark provides a controlled evaluation environment to understand whether the multimodal reasoning capabilities developed through our teacher aggregation methodology can effectively generalize to audio-only inference tasks.

Table 3: Comparison of audio-visual reasoning models on benchmark datasets. DailyOmni, OmniBench, and MMAR are tested in zero-shot mode without further finetuning. AVQA results are considered fine-tuned since the training dataset is derived from AVQA's training set. We report reproduced baseline results for Qwen2.5 Omni marked with $^*$.

| Model | # Params | Reasoning | DailyOmni (Video+Audio) | OmniBench (Image+Audio) | AVQA$^\dagger$ (Video+Audio) | MMAR (Audio-Only) |
|---|---|---|---|---|---|---|
| *7B Audio-Visual Models* | | | | | | |
| Qwen2.5 Omni (Xu et al. (2025)) | 7B | × | 44.0 | 44.2 | - | 56.7 |
| Qwen2.5 Omni$^*$ (Xu et al. (2025)) | 7B | × | 51.5 | 50.7 | 84.9 | 56.5 |
| Echolnk (Xing et al. (2025)) | 7B | ✓ | 46.2 | 46.5 | - | - |
| Omni-R1 (Zhong et al. (2025)) | 7B | ✓ | 46.8 | 46.9 | - | - |
| HumanOmni (Zhao et al. (2025)) | 7B | ✓ | 47.6 | 44.9 | - | - |
| Ola-7B (Liu et al. (2025b)) | 7B | × | 52.3 | 45.3 | - | - |
| AV-Reasoner (Lu et al. (2025)) | 7B | ✓ | **53.8** | 48.3 | - | - |
| AVATAR (Kulkarni & Fazli (2025)) | 7B | ✓ | 47.0 | **49.1** | - | - |
| *Modality-Specific Teachers* | | | | | | |
| Kimi-VL-Thinking (Team et al. (2025)) | - | ✓ | - | 33.5 | - | *N/A* |
| AF3 *(think)* (Goel et al. (2025)) | - | ✓ | - | 28.9 | - | 60.1 |
| *3B Audio-Visual Models* | | | | | | |
| Qwen2.5 Omni (Xu et al. (2025)) | 3B | × | 42.9 | 42.4 | 87.9 | 53.8 |
| Qwen2.5 Omni$^*$ (Xu et al. (2025)) | 3B | × | 43.1 | 50.2 | 88.3 | 53.7 |
| AVATAR (Kulkarni & Fazli (2025)) | 3B | ✓ | 44.7 (+1.8) | 45.8 (+3.4) | - | - |
| **Ours** | 3B | ✓ | **45.5** (+2.4) | **54.7** (+4.5) | **90.1** (+1.8) | **56.2** (+2.5) |

## 4.2 IMPLEMENTATION DETAILS

For all experiments, we use Qwen2.5-Omni-3B (Xu et al. (2025)) as base student model with frozen vision and audio modules. Fully supervised fine-tuning is conducted on $18,279$ samples over $1$ epoch with an effective batch size of 32 (1 sample per device $\times$ 8 gradient accumulation steps $\times$ 4 H100 GPUs). We use a learning rate of $2e-6$ with cosine scheduling, AdamW optimizer ($\beta_1 = 0.9, \beta_2 = 0.999, \epsilon = 1e-8$), weight decay of 0.01, and 100 warmup steps. Training employs DeepSpeed ZeRO Stage 2 optimization with CPU offloading and bfloat16 precision. For reinforcement learning, we use identical infrastructure with GRPO-specific hyperparameters: group size $G = 4$, clipping parameter $\epsilon = 0.2$, KL regularization coefficient $\beta = 0.01$, and temperature 1.

## 4.3 COMPARISON TO STATE-OF-THE-ART

Table 3 shows the performance of the model trained with the proposed approach against existing audio-visual reasoning models across four benchmark datasets.

The resulting 3B parameter model achieves strong performance, both in terms of absolute accuracy and relative improvement over our reproduced baseline. On DailyOmni, the model achieves $45.5\%$ accuracy, outperforming both 3B baselines and remaining competitive with 7B models.

On OmniBench, our model reaches $54.7\%$ accuracy. Our model improves upon our reproduced baseline by $+4.5$ percentage points ($54.7\%$ vs. $50.2\%$), whereas AVATAR-3B's reported gain over their baseline is $+3.4$ points ($45.8\%$ vs. $42.4\%$). Note that we were unable to replicate the accuracy reported by Kulkarni & Fazli (2025) due to unavailable code, we instead compare the relative gains over our own Qwen2.5-Omni-3B baseline.

On AVQA, the model achieves $90.1\%$ accuracy, representing a $+1.8$ percentage point improvement over the reproduced Qwen2.5-Omni-3B baseline ($90.1\%$ vs. $88.3\%$). While this improvement is more modest compared to other benchmarks, it is important to note that AVQA represents a fine-tuned evaluation scenario since the training dataset is derived from AVQA's training set.

For MMAR, an audio-only reasoning benchmark, the model achieves $56.2\%$ accuracy, outperforming the baseline by $+2.5$ percentage points ($56.2\%$ vs. $53.7\%$). The improvement on MMAR validates that the audio reasoning capabilities developed through the teacher aggregation methodology generalize effectively beyond the multimodal training domain approaching that of the specialized audio teacher AF3 *(think)* ($60.1\%$), suggesting that multimodal training with aggregated reasoning chains can develop strong single-modality capabilities as a byproduct of audio-visual learning.

Overall, the results show that reasoning-capable models outperform their non-reasoning counterparts across parameter sizes, validating the importance of structured reasoning in audio-visual tasks.

Table 4: Impact of training stages and evaluation on single modality performance. We evaluate using only audio (A), only vision (V), and both modalities (AV) on OmniBench.

| Model | SFT | RL | Mod. | OmniBench |
|---|---|---|---|---|
| Qwen2.5-Omni 3B | × | × | A | 39.4 |
| | × | × | V | 42.7 |
| | × | × | AV | 50.2 |
| Qwen2.5-Omni 3B (RL Only) | × | ✓ | A | 40.5 |
| | × | ✓ | V | 43.1 |
| | × | ✓ | AV | 53.0 |
| **Ours** | ✓ | ✓ | A | 40.7 |
| | ✓ | ✓ | V | 44.3 |
| | ✓ | ✓ | AV | **54.7** |

Table 5: Ablation on merger models. Using an LLM merger with the same backbone as the student model enhances performance.

| LLM Merger Model | Student Model | OmniBench |
|---|---|---|
| Gemma3-12B-It | Qwen2.5-Omni-3B | 48.1 |
| Qwen2.5-14B-Instruct | Qwen2.5-Omni-3B | **54.7** |

Table 6: Performance on different OmniBench difficulty subsets. Our model outperforms the base model on all difficulty levels.

| Model | Easy | Medium | Hard |
|---|---|---|---|
| Qwen2.5-Omni | 62.5 | 46.5 | 28.9 |
| **Ours** | **72.9** | **50.7** | **32.8** |

## 4.4 ABLATION STUDIES

**Evaluation of SFT fine-tuning.** We first asses the impact of the supervised fine-tuning step on the generated reasoning traces compared to the Qwen2.5-Omni 3B baseline, as well as to the same model trained only with an RL objective. As shown in Table 4, simply training the model with an RL objective leads to an improvement of $2.8\%$ which is consistent with improvements reported by other approaches (see Table 3). It further shows that the proposed 2-stage training with a SFT cold-start phase based on the generated reasoning traces is able to improve over this baseline by another $1.7\%$.

**Single Modality Performance.** To investigate the impact of training exclusively on multimodal data, we evaluate the model using only one modality at a time. As shown in Table 4, the proposed training strategy also leads to a modest improvement in the single-modality settings: compared to the Qwen2.5-Omni 3B baseline, our model achieves a $+1.3$ point improvement in the audio-only setting ($40.7\%$ vs. $39.4\%$) and a $+1.6$ point improvement in the vision-only setting ($44.3\%$ vs. $42.7\%$). This can be considered as an indication for reasoning transfer learning as the SFT dataset is composed mainly of questions that require both audio and vision ($99.0\%$) (see Appendix Table 7) and as both our supervised fine-tuning (SFT) with reasoning chains plus RL and the RL-only baseline are trained solely on audiovisual inputs, without any modality dropout or single-modality augmentation.

**Different merger models.** We investigate the impact of using different teacher models for cross-modal aggregation in our pipeline. As shown in Table 5, we compare two merger models: Gemma3-12B-It and Qwen2.5-14B-Instruct. The results show a substantial performance difference, with Qwen2.5-14B-Instruct achieving 54.7% accuracy on OmniBench compared to 48.1% for Gemma3-12B-It. Notably, during training, we observe that the model fine-tuned with reasoning chains generated by Qwen2.5-14B-Instruct converged significantly faster to the multiple-choice question (MCQ) format compared to the Gemma3-based merger. This suggests that using a teacher model from the same architectural family as the student model facilitates more efficient knowledge transfer, as the student model does not need to adapt to a substantially different token distribution during fine-tuning.

**Omnibench Subsets.** To further analyses the impact of the reasoning capabilities, we categorize OmniBench questions into difficulty subsets based on teacher model performance: *easy* (both teachers correct, 64 questions), *medium* (one teacher correct, 456 questions), and *hard* (neither correct, 619 questions). As shown in Table 6, our model outperforms the Qwen2.5-Omni baseline across all difficulty levels: easy (72.9% vs. 62.5%, +10.4 points), medium (50.7% vs. 46.5%, +4.2 points), and hard (32.8% vs. 28.9%, +3.9 points). The *easy* subset shows the largest improvement but exhibits greater variability due to its smaller sample size. The predominance of hard questions (54%) demonstrates that OmniBench is well-suited for evaluating cross-modal reasoning.

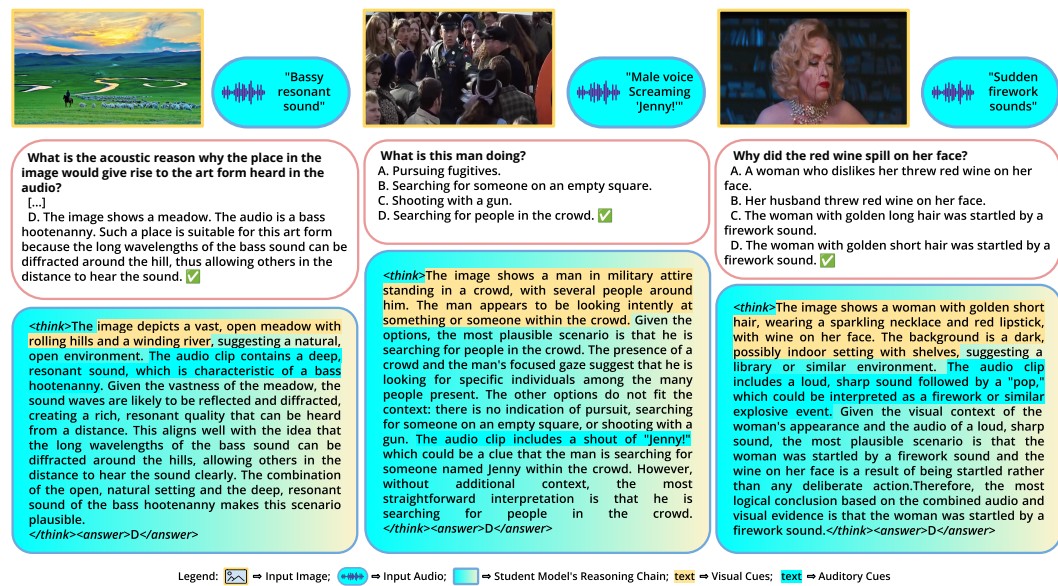

Figure 2: Qualitative results of the AVRT-trained model on OmniBench. The model trained on our collected AVRT dataset is able to integrate audio and visual content to answer the question and generate high-quality reasoning chains. *Best viewed in color and zoomed in.*

## 4.5 QUALITATIVE RESULTS

We finally provide qualitative results of reasoning chains produced by the student model after fine-tuning on the proposed AVRT reasoning traces on OmniBench in Fig. 2. The results show that with only SFT on the generated reasoning chains, the model is able to reason about the image and audio content to answer the question. In the first example, the model correctly associates the acoustic properties of a bass "hootenanny" with the open meadow environment, demonstrating understanding of how sound propagates differently in open versus enclosed spaces. The second example showcases more sophisticated multi-modal reasoning, where the model uses visual cues (people around the main character) to contextualize the audio (male voice calling "Jenny!") and correctly identifies the scenario as searching for people in a crowd among multiple plausible options. The third example illustrates the model's ability to connect temporal audio events (firework sounds) with visual evidence (wine spill on face). As shown in the figure, the model learns to incorporate both visual and auditory cues to arrive at correct answers. These results are on OmniBench, which is a particularly challenging dataset, and demonstrate the model's ability to generalize from its training domain (AVQA videos with 8 frames and audio) to a different evaluation domain (single image and audio inputs).

## 5 CONCLUSION

We introduced AVRT, a novel framework that generates high-quality audio-visual reasoning data by distilling knowledge from specialized single-modality teachers, enabling effective supervised fine-tuning of student models with minimal reinforcement learning post-training. To this end, the pipeline uses two specialized reasoning teachers, one for audio and one for vision, to extract expert reasoning traces of each modality separately. The resulting traces are then merged and formatted by an LLM merger model into a single, multimodal reasoning trace. We then use those reasoning traces as cold-start in a two stage training pipeline. The resulting 3B parameter model achieves state-of-the-art performance: 54.7% on OmniBench, 45.5% on DailyOmni, and 56.2% on MMAR, establishing new benchmarks for efficient audio-visual understanding.

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

# A   APPENDIX

## A.1   ADDITIONAL DATASET STATISTICS

Table 7: Additional statistics for the AVRT-20K dataset showing answer option distribution and question modality requirements.

| Answer Option | Train / Val (%) |
| --- | --- |
| A | 24.5 / 27.1 |
| B | 25.2 / 27.3 |
| C | 24.9 / 23.3 |
| D | 25.4 / 22.3 |

| Question Relations | Train / Val (%) |
| --- | --- |
| Both (Audio + Visual) | 99.0 / 98.9 |
| Sound Only | 0.7 / 0.7 |
| Visual Only | 0.3 / 0.3 |

The AVRT-20K dataset has balanced answer choices with each option (A-D) appearing roughly 25% of the time. Nearly all questions (99%) require both audio and visual information to answer correctly, with less than 1% being answerable using only one modality.

## A.2   PROMPTS

In this section, we provide the prompts used in this work for both teacher models (Kimi-VL-Thinking and Audio Flamingo 3 (*think*)), the merger model (Qwen2.5-14B-Instruct), and the student model (Qwen2.5-Omni-3B).

### A.2.1   VISUAL TEACHER PROMPT

> **Visual Teacher (Kimi-VL-Thinking) Prompt**
>
> You are an intelligent vision agent. I will provide you with 8 representative frames from a video (evenly distributed across the video duration) and a question about the video content in MCQ format. You need to first provide a thorough description of what you're seeing across these video frames, then add Chain-of-Thought-type reasoning to analyze the visual content, and finally provide your answer. Here is an example:
> Input Question: What type of activity is happening in this video? Choose one among the following options:(A) Crime thriller scene (B) Documentary narration\n(C) Romantic comedy scene\n(D) Action movie or racing scene\n
> Expected response format:\n
> Visual Description: Across these video frames, I can see a progression of high-speed chase scenes with vehicles moving rapidly through an urban environment. The frames show consistent dynamic motion, intense lighting, and what appears to be an ongoing action sequence with cars and possibly motorcycles. The temporal progression across frames reveals the continuous high-energy nature of the content.\n
> Reasoning: Based on the consistent high-speed vehicle movement visible across multiple frames, the sustained dynamic camera work, intense lighting throughout the sequence, and the overall action-oriented visual elements that persist across the video timeline, this content would be most suitable for action-focused scenarios that require high-energy sequences. The visual elements strongly suggest this is an action movie or racing scene rather than other genres like crime thriller, documentary, or romantic comedy.\n
> Answer: (D) Action movie or racing scene\n
> Follow this format: provide a detailed visual description analyzing the temporal progression across frames, then your reasoning considering the full video context and evaluating each option, then the final answer. For answers that do not require complex reasoning (e.g., for a question like "What color is the object?" or "How many people are in the image?" where the answer is direct), still provide the visual description but keep the reasoning brief.\n
> Here is the input question:

### A.2.2   AUDIO TEACHER PROMPT

We use the same prompts that are available in the Audio Flamingo 3 paper for their teacher models during the training pipeline. We adopted these prompts to ensure optimal performance from the

model and generate descriptive reasoning chains that maintain consistency with the original model's training methodology.

---

**Audio Teacher (Audio Flamingo 3 *think*) Prompt**

You are an intelligent audio agent. I will provide you with an audio and a question about the audio in MCQ format. You need to first provide a thorough description of what you're hearing in the audio, then add Chain-of-Thought-type reasoning to analyze the audio content and evaluate each option, and finally provide your answer. Here is an example:

Input Question: What type of soundtrack would this piece be most suitable for? Choose one among the following options:(A) Crime thriller movie (B) Documentary narration\n(C) Romantic comedy movie\n(D) Futuristic movie or car racing video game\n

Expected response format:\n

Audio Description: This audio features a high-energy electronic track with a driving beat, synthesized sounds, and confident rap vocals. The lyrics mention themes of speed and success, including phrases like 'living automatic' and references to new cars. The production has a modern, polished sound with heavy use of electronic elements.\n

Reasoning: Based on the driving beat, confident rap vocals, mentions of speed and success, and overall high-energy modern production with electronic elements, this piece would be most suitable for high-octane, modern scenarios that require energetic background music. Evaluating the options: (A) Crime thriller movies typically use more suspenseful, darker soundtracks; (B) Documentary narration usually requires more neutral, informative background music; (C) Romantic comedy movies generally feature lighter, more melodic soundtracks; (D) Futuristic movies or car racing video games would benefit from exactly this type of high-energy electronic music with themes of speed and technology.\n

Answer: (D) Futuristic movie or car racing video game\n

Follow this format: provide a detailed audio description first, then your reasoning that evaluates each option, then the final answer. For answers that do not require complex reasoning (e.g., for a question like "Who performs the vocals in this song?" or "What primary instrument is featured in this piece?" where the answer is direct), still provide the audio description but keep the reasoning brief.\n

Here is the input question:

---

### A.2.3 MERGER PROMPT

---

**Merger (Qwen2.5-14B-Instruct) Prompt**

You are an intelligent multimodal agent. I will provide you with a question in MCQ format, along with separate audio and visual analyses from specialized models. Your task is to merge these analyses into a coherent reasoning chain that integrates both modalities to arrive at the correct answer.

Question: {question}{formatted_choices}

Correct Answer: {correct_answer}

Audio Analysis: {audio_reasoning}

Visual Analysis: {vision_reasoning}

Instructions:

- Don't acknowledge that you already know the answer!

- Act as if you generated the reasoning and then you came across the right answer by yourself!

- Write plain English, but this time, format your merged reasoning inside <think>... </think>

- At the end, output your final answer (just the letter, e.g., A, B, C, or D) inside <answer>... </answer>

- Write sentences that integrate both audio and visual evidence

- Explain how the audio and visual clues work together to lead you to the conclusion

- Make the explanation thorough but succint

Combined Analysis:

---

