# OpenReview forum: "AVRT: Audio-Visual Reasoning Transfer through Single-Modality Teachers"
_ICLR.cc/2026/Conference — ICLR 2026 Conference Withdrawn Submission_

### Official Review · Reviewer_Ko9m · 2025-10-23

**Soundness:** 2
**Presentation:** 2
**Contribution:** 2
**Rating:** 2
**Confidence:** 4

**Summary:**

This paper proposes a way to create audio-visual reasoning data from audio-only and visual-only models. The authors show the data created is helpful to achieve better performance in reasoning in a somewhat limited setup.

**Strengths:**

1. The data creating pipeline is clear and the paper is easy to follow.
2. Delving into better SFT data for cold-starting audio-visual LLM reasoning is a useful direction.

**Weaknesses:**

1. The completeness of the evaluation:
The authors have listed a range of 7B models, whereas only showing results for 3B models using their data. Also the author should compare to some reasoning-oriented benchmark such as RivaBench [1].
[1] Sun et al. "video-SALMONN-o1: Reasoning-enhanced Audio-visual Large Language Model". Proc. ICML. 2025.

2. Missing ablation studies:
A core contribution of the dataset is to use a merger to merge audio and visual-only reasoning for SFT so it would be useful to see how the model performs when only SFT stage is used, and an analysis of how this affects the later RL stage should be given.

3. Contribution seems limited:
From a perspective of someone who have hands-on experience of creating SFT training data for reasoning, I do not feel such a data creation pipeline worth publishing, unless some unexpected conclusions or findings are provided, together with strong theoretical grounding. I did not find either in this paper.

**Questions:**

Have the author tried actual or deeper distillation from audio-only or visual-only teachers? Not just creating data.

---

### Official Review · Reviewer_Rz8Y · 2025-10-23

**Soundness:** 2
**Presentation:** 2
**Contribution:** 2
**Rating:** 4
**Confidence:** 3

**Summary:**

The paper proposes a pipeline for generating synthetic reasoning traces over audio. The approach first processes joint audio–video data with modality-specific reasoning models, producing separate audio and video reasoning traces. These traces are then merged by an LLM into a unified reasoning trace, resulting in audio–video reasoning data. Using this data, the authors fine-tune 3B-parameter models with a combination of supervised finetuning and reinforcement learning, achieving performance comparable to larger 7B-parameter models.

**Strengths:**

- The authors introduce a dataset that has clear potential value for the community, and their experiments demonstrate consistent performance gains across multiple benchmarks relative to the base model.
- The results reinforce prior findings [(Lu et al., 2025), (Kulkarni & Fazli, 2025)] that reasoning traces can enhance the performance of audio–video models.
- In contrast to earlier approaches [(Lu et al., 2025), (Kulkarni & Fazli, 2025), (Goel et al., 2025)], the proposed reasoning traces are more natural, as they integrate both audio and video, rather than relying on separate modality-specific reasoning [(Kulkarni & Fazli, 2025)] or exclusively on LLM-generated reasoning [(Lu et al., 2025)].
- An interesting aspect of the experiments is that supervised finetuning on the dataset prior to reinforcement learning further boosts performance.

**Weaknesses:**

- The central contribution lies in extracting reasoning signals independently from audio and video and then merging them into a single reasoning trace. However, there is no ablation study isolating the impact of reasoning traces from a single modality. The closest analysis appears in Table 4, but my understanding is that the evaluated model there is trained on full audio–visual reasoning traces rather than on modality-specific traces.
- In Table 3, the comparison between the final row and Qwen2.5 Omni* (the base model) does not demonstrate statistically significant improvements, making it difficult to justify the method’s effectiveness.
- The paper highlights that supervised finetuning on reasoning traces provides a stronger starting point for reinforcement learning (in the abstract, introduction, and Figure 1), yet offers little analysis of why this occurs. For example, does SFT make it easier for the RL agent to obtain positive feedback?

**Questions:**

- It is unclear whether the proposed setup actually improves performance. In Table 3, the reported gains appear minimal, if present at all.
- Independent of overall performance, reasoning traces could be an important contribution. Are there specific benefits obtained by combining reasoning from both modalities, as opposed to using each one individually?
- Do you have insights into why supervised finetuning provides a better cold start for the reinforcement learning stage?

---

### Official Review · Reviewer_rCKC · 2025-10-30

**Soundness:** 2
**Presentation:** 3
**Contribution:** 2
**Rating:** 2
**Confidence:** 5

**Summary:**

The paper proposes AVRT, a pipeline that builds an audio‑visual reasoning dataset by distilling chain‑of‑thought (CoT) traces from two single‑modality teachers: an audio teacher and a visual teacher and then merging the two traces with a text‑only LLM merger. The merged trace is used to SFT a student model with vision/audio encoders frozen, followed by GRPO with a simple reward on output format and multiple‑choice answer accuracy. The authors construct AVRT‑20K dataset (approx. 18.3k train and 945 val). With the integration of proposed reasoning chains in the AV modes, the authors report performance gains.

**Strengths:**

- The paper is clearly structured, with an informative pipeline diagram.
- Concise and Readable sections and tables.
- Simple and modular recipe for building AV reasoning traces without multimodal annotation.
- The proposed dataset could be used to train the AV models in future,

**Weaknesses:**

- Most scores in  Table 3 of main paper are empty which makes it hard to establish the effectiveness of the proposed method.
- It seems that there is data leakage in the training. The merger LLM already sees the gold label. The merger prompt includes the correct answer, which undermines the central claim about the “composed reasoning.”
- Filtering to reasoning traces where both teachers are correct induces data bias. By keeping only items where both teachers are correct the dataset may over‑represent easy/teacher‑friendly cases and omit the conflict‑resolution scenarios most indicative of cross‑modal reasoning benefits. It is worth checking if the same reasoning traces are filtered out using other teachers and derive some insights out of it.
- The comparison scope is limited. Main comparisons rely on a reproduced Qwen2.5‑Omni‑3B baseline and reported numbers for several 7B systems.
- Training uses 8 frames per video and 10‑second audio for teachers; it would help to analyze how such input pruning affects temporal reasoning and transfer to single‑image + audio setups like OmniBench.
- Apart from the SFT, the proposed method has lot of similarities with the recent work "AURELIA" [1] which distills reasoning at the test-time. The paper is not cited by the authors although being very relevant. How does your method compare with AURELIA?
- Overall the merger’s access to the gold answer and the both‑teachers‑correct filtering substantially weaken the claim that AVRT constructs high‑quality, label‑agnostic cross‑modal reasoning traces


References:

[1] Sanjoy Chowdhury, Hanan Gani, ..et al. AURELIA : Test‑time Reasoning Distillation in Audio‑Visual LLMs. ICCV 2025.

**Questions:**

- Do the Table 4 results include the Reasoning chain with the query when SFT and RL are not used? If not, what  happens when we simply pass the reasoning chain with the query without SFT? Also what happens when passing the reasoning chain with query without both SFT and RL (kind of zero-shot)?
- Do traces stay consistent if you mask either modality at inference? Any counterfactual tests to check whether the CoT actually uses both streams.

---

### Note · Authors · 2025-11-13

I have read and agree with the venue's withdrawal policy on behalf of myself and my co-authors.